# Patients' and physiotherapists' perspectives on implementing a tailored stratified treatment approach for low back pain in Nigeria: a qualitative study

Mishael Adje ,[1,2] Jost Steinhäuser,[2] Kay Stevenson,[3] Chidozie Emmanuel Mbada,[4] Sven Karstens[5]

¹Therapeutic Sciences, Trier University of Applied Sciences, Trier, Germany
²Institute of Family Medicine, University of Lübeck, Lübeck, Germany
³The Impact Accelerator Unit, The Medical School Keele University, Keele, UK
⁴Department of Medical Rehabilitation, Obafemi Awolowo University, Ile-Ife, Nigeria
⁵Therapeutic Sciences, Department of Computer Science, Trier University of Applied Sciences, Trier, Germany

**Correspondence to**
Dr Mishael Adje;
adjem@hochschule-trier.de

## ABSTRACT

**Background** Stratified care has the potential to be efficient in addressing the physical and psychosocial components of low back pain (LBP) and optimise treatment outcomes essential in low-income countries. This study aimed to investigate the perceptions of physiotherapists and patients in Nigeria towards stratified care for the treatment of LBP, exploring barriers and enablers to implementation.

**Methods** A qualitative design with semistructured individual telephone interviews for physiotherapists and patients with LBP comprising research evidence and information on stratified care was adopted. Preceding the interviews, patients completed the Subgroups for Targeted Treatment tool. The interviews were recorded, transcribed and analysed following grounded theory methodology.

**Results** Twelve physiotherapists and 13 patients with LBP participated in the study (11 female, mean age 42.8 (SD 11.47) years). Seven key categories emerged: *recognising the need for change, acceptance of innovation, resistance to change, adapting practice, patient's learning journey, trusting the therapist* and *needing conviction.* Physiotherapists perceived stratified care to be a familiar approach based on their background training. The prevalent treatment tradition and the patient expectations were seen as major barriers to implementation of stratified care by the physiotherapists. Patients see themselves as more informed than therapists realise, yet they need conviction through communication and education to cooperate with their therapist using this approach. Viable facilitators were also identified as patients' trust in the physiotherapist and adaptations in terms of training and modification of the approach to enhance its use.

**Conclusion** Key barriers identified are the patients' treatment expectations and physiotherapists' adherence to the tradition of practice. Physiotherapists might facilitate implementation of the stratified care by communication, hierarchical implementation and utilisation of patients' trust. Possibilities to develop a consensus on key strategies to overcome barriers and on utilisation of facilitators should be tested in future research.

## STRENGTHS AND LIMITATIONS OF THIS STUDY

⇒ The barriers and enablers to implementation of stratified care in low-income settings like Nigeria are explored from the perspective of patients and physiotherapists in this study.
⇒ This study adhered to a rigorous circular grounded theory method to systematically draw out comparisons and variations in perspectives of 25 participants until saturation was reached.
⇒ Participants were motivated to engage in productive discussions and involved in co-designing introductory materials and tailoring interview guides.
⇒ The researchers' own subjectivity, beliefs and experiences might have played a role in shaping the outcome, but this can be seen as an inevitable and integral part of using grounded theory.
⇒ The need for stakeholder awareness in areas of divergent perspectives, to inform quality communication and patient conviction seen in this study, is in concordance with current evidence on stratified care implementation in other contexts.

physiotherapy clinics in Nigeria.[1–3] It is a prominent cause of disability and the effects have been immense in terms of cost, time and productivity loss.[4–7]

In recent decades, there has been remodelling in the management of LBP. Current literature corroborated by multiple guidelines recommends the biopsychosocial model of care.[8–13] These resources highlight the prognostic significance of psychosocial factors such as depression, fear-avoidance behaviours, catastrophising, distress among many individual patients and their culpability in worsening musculoskeletal conditions.[13 14]

Stratified care encompasses specific concepts in the management of LBP, differentiating and targeting prognostic subgroups, and aligning the risk of an unfavourable treatment outcome with specific evidence-based treatment procedures.[15] Its

## INTRODUCTION

Low back pain (LBP) is among the most frequent reasons for visits to outpatient

principles lean towards prognosis, de-emphasising pain, disputing unhelpful beliefs and equipping patients on self-management strategies.[8 9 13] It aims to guide clinicians to identify psychosocial risk factors hindering patient recovery and fast track accurate treatment decisions.[16 17]

One well-known stratification approach with demonstrated effectiveness in both clinical and cost outcomes is the Subgroups for Targeted Treatment (STarT-Back) approach.[18 19] It involves categorising patients with LBP into low, medium and high risk categories using the STarT-Back tool (SBT) and allocating matched treatments to participants in each category.[20]

The SBT was developed for primary healthcare settings; it is a validated tool that can be administered paper based or online based. It consists of nine items, designed to assess modifiable risk factors.[20] The first four questions address aspects of the patient's physical characteristics asking about the spread of the pain towards the leg, to the neck or shoulders, and disability in dressing or walking. The next five questions comprise the psychosocial subscale relating to patients' psychological characteristics, including fear of movement, anxiety, catastrophising, depression and bothersomeness.[20]

In a large-scale study evaluating the effectiveness of the STarT-Back approach, the score from the SBT was used to classify patients into three subgroups.[19] Depending on the subgroup, there are treatment recommendations. If a patient scores between 0 and 3 in total on the SBT, they are allocated to the low-risk subgroup. These comprise patients who will improve with little or no intervention. They are helped with advice and pain medication. If patients score between 4 and 9 on the grouping tool but have 3 or fewer of the five subscales, they are allocated to the medium-risk subgroup. They have a moderate likelihood of poor treatment outcomes, and they have fewer physical and psychological components of pain. The emphasis here is on physiotherapy and self-management.[21] Patients who score 4 of 5 on the psychological subscale are allocated to the high-risk subgroup. These patients possess complex prognostic factors impeding recovery and need more care due to these. Such care comprises psychologically informed physiotherapy, aimed to resolve physical and psychosocial components of pain.[15 19] All target treatments are aimed to reduce disability, reduce pain where possible, improve psychological functioning and enable the patient to manage ongoing and/or future episodes of back pain.[21]

Studies from the UK and New Zealand reveal that the implementation of the STarT-Back model of care has the potential to reduce disability, save costs, improve patient satisfaction, reduce the number of patient visits and increase clinicians' competence,[19 22 23] except for a US study indicating no effect on patient outcomes.[24] A recent systematic review shows that this approach provides benefits in terms of clinical outcomes and health-related cost savings for all three subgroups of patients, and patients in the high-risk subgroup who received psychologically informed treatment had significant improvements in pain, disability, depression and general health.[25] The STarT-Back approach is described as best practice in the UK international guideline for the management of LBP[11 26] and has been recommended from studies in the UK and other countries to be efficacious and have a positive impact on LBP outcomes.[18 19 21] However, these are mostly high-income countries (UK, Denmark) with financially sound healthcare systems, thus further care is taken to ensure proper implementation in countries in Africa.

Studies on implementation efforts in higher income countries exhibit challenges further amplified when considering contextual factors present in lower/middle-income countries.[15 18 19] A study shows that socioeconomic status is an important treatment effect modifier when using the STarT-Back approach.[27] Differences in settings can also play a major role in determining the outcome of implementation.[24] Unique circumstances make it challenging for some physiotherapists to consider practice guideline recommendations when planning treatment.[28] This leads to the omission of psychosocial risk factors, which undermine treatment outcome[29] and are present in over 60% of patients with chronic LBP in Nigeria.[30] In combating this, implementation of the STarT-Back model of care promises a viable solution. Since there is no universal instruction manual for implementation, it is recommended that stratified care for LBP will first need adaptation in terms of setting before wider implementation is considered. The first step to this is identifying the contextual barriers and enablers to implementation.[15]

This study was aimed to explore the perspectives of physiotherapists and patients with LBP, identifying perceived barriers and enablers to the implementation of stratified care in Nigeria.

## METHODS
A qualitative design was adopted in this study. Semistructured individual telephone interviews for patients and physiotherapists were extensively executed following grounded theory methodology.[31 32] The Consolidated Criteria for Reporting Qualitative Research was employed for the study report.[33]

### Interviewer/facilitator
The research team was made up of five researchers: MA (male), a Nigerian physiotherapist with experience in musculoskeletal health; SK (male), a physiotherapist with experience in qualitative and health services research; JS (male), a general practitioner with experience in medicine and qualitative research; KS (female), a physiotherapist with experience in health promotion and implementation of the STarT-Back approach; and CEM (male), a physiotherapist with experience in musculoskeletal research in Nigeria. The interview guidelines were prepared and reviewed by MA, SK and JS, and the interviews were conducted by MA having no direct relationship with the participants.[34]

## Participant recruitment

Participants were theoretically sampled for the interview rounds.[32 35] Participants were contacted following set inclusion criteria specifically comprising physiotherapists and patients living in Nigeria.

To be included in this study, physiotherapist participants had to be licensed and registered by the regulating body (Medical Rehabilitation Therapist Board) of Nigeria and practising physiotherapy in Nigeria with entry-level qualification and above. They should have worked on the STarT-Back information on the introductory video provided. Patient participants had to be diagnosed as having non-specific LBP by their physician or physiotherapist and have visited a healthcare institution for physiotherapy care. They should speak and understand the English language fluently and should be able to fill the STarT-Back questionnaire. All participants must be above 18 years.

Patients with a diagnosis or having any signs or symptoms pointing to a severe disease condition (eg, cancer) as the cause of their LBP were excluded from this study.

Physiotherapist participants were from diverse specialties in diverse work settings, with varying work experience and levels of training. Patients who had LBP of varying severity and any episodic duration spread across the three risk subgroups in the STarT-Back classification and were receiving treatment from any health institution. They were approached and invited by the physiotherapists who have agreed to participate.[36]

Sociodemographic details comprising age, sex and work status were obtained from participants. The STarT-Back risk categories were determined. Physiotherapist participants provided details on age, sex, previous training, working experience (overall/with patients with LBP), qualification and work setting. Participants were recruited until saturation was reached.[37]

For the introduction of the concept of stratified care to patients and physiotherapists, interactive videos were prepared and presented before the interviews.[19] The videos were designed with a voice layover and interactive questions embedded within to keep participants involved and to help them understand key aspects. These videos were prepared following the guidelines for instructional videos by Norman[38] with input from the SBT development group in the UK. The content was developed for two specialised videos for the participating physiotherapist and one video for the patient with underlying principles from Main et al[39] as shown in table 1. Patients' video was co-produced with patient input, using lay understandable language, and physiotherapists' videos were co-created with input from physiotherapists. Stakeholder input informed the content, timing, use of language and length of presentation.[40]

## Data collection

Interviews were conducted over the phone at the convenience of the participants.

**Table 1** Content of the introductory presentation adapted from Main et al[39]

| For patients | For physiotherapists |
|---|---|
| The rationale for using SB approach, including potential clinical benefits | The rationale for SB approach |
| Psychosocial components of LBP | Psychosocial barriers to progress |
| Description of the SB approach | Grouping using the SB tool |
| ► Content and purpose | ► Content and purpose |
| ► Definition of subgroups | ► Definition of subgroups |
| ► Matched treatments | How to use and score the tool |
| Treatment outcome using the approach | Scientific underpinning of the approach |
| | Clinical and economic benefits |

LBP, low back pain; SB, STarT-Back; STarT-Back, Subgroups for Targeted Treatment.

The interview guides for patients and physiotherapists were prepared in four phases: brainstorming, collection, sorting, examining questions and consultations as described in research.[41 42] In developing the initial interview questions, the aspect of implementation was duly considered.[43]

### Patient guideline

Developed questions were rephrased to simpler forms for patients and to reveal the patient's perspective (online supplemental table 1), then sorted repeatedly based on categories,[44] allowing for open but thematically structured interview questions.[45] They were then reviewed by a patient with LBP and modified by inputs.[40] Four key questions were developed with additional maintenance questions and follow-up questions for patients with LBP.

### Physiotherapist guideline

The questions for physiotherapists were rephrased to reflect physiotherapists' perspectives (online supplemental table 2). Two physiotherapists with experience in qualitative interviewing reviewed the questions and made inputs. These were adopted into the guideline ready for use. Four key questions were developed with additional maintenance questions and follow-up questions for physiotherapists.

A pilot test of the interview guidelines was carried out on a subsample of the target population (n=5) (three physiotherapists and two patients with LBP). Verbalisation feedback by participants informed the modification of the interview questions and approach strategy, as exemplified in a study by Pepper et al.[46] The focus of the interviews was hence modified simultaneously with theoretical sampling.[40]

The telephone interviews were conducted by MA following the pattern of a problem-centred interview.[45] The interviews were tape-recorded and handwritten notes were taken during the interviews. Verbal consent was received from each participant before commencement. Questions were flexibly rearranged to express a dialogue and improve the flow of the interview, beginning with

**Table 2** Patients' sample description

| Set* | Title† | Age group (years) | Sex | Work status | STarT-Back classification‡ |
|------|--------|-------------------|-----|-------------|----------------------------|
| 1 | Pat 1 | >50–60 | M | Paid work | High risk |
| 2 | Pat 2 | >50–60 | M | Paid work | Low risk |
| | Pat 3 | >40–50 | M | Paid work | High risk |
| | Pat 4 | >70–80 | M | Retired | High risk |
| | Pat 5 | >60–70 | M | Retired | Low risk |
| 3 | Pat 6 | >60–70 | F | Self-employed | Medium risk |
| | Pat 7 | >30–40 | F | Self-employed | Medium risk |
| | Pat 8 | >40–50 | M | Paid work | High risk |
| | Pat 9 | >40–50 | F | Self-employed | High risk |
| | Pat 10 | >50–60 | F | Self-employed | Medium risk |
| 4 | Pat 11 | >30–40 | F | Paid work | High risk |
| | Pat 12 | >40–50 | M | Paid work | Medium risk |
| | Pat 13 | >30–40 | M | Self-employed | High risk |

*First, second, third and fourth iterative interview rounds.
†Patients' pseudonymous designation comprising the numerical order of interviews used to reference patients' quotes.
‡Based on the STarT-Back tool. Patients who score 0–3 were allocated to the low-risk subgroup, 4–9 but 3 or fewer of the five subscales are medium risk, 4 of 5 on the psychological subscale are high risk. Higher scores indicate an increasing complexity of the condition.
Pat, patient; STarT-Back, Subgroups for Targeted Treatment.

warm-up questions. These questions drew insight into the expert's background to create a context and relaxing warm-up atmosphere.

Notes were summarised back to the participants after each interview and memos were written about each interview. This process was repeated for the various rounds until saturation, where no new categories emerged.[37]

### Data analysis

A simple transcription method as described by Dresing and Pehl was used[47] with the help of Easy transcript software.[48] Pseudonyms were used in each transcript making it recognisable during the coding process and differences between patients and physiotherapists. Transcripts were plain text only to facilitate the coding process and were read separately by a second researcher with experience in qualitative analysis.

### Coding

Transcripts were read and reread by MA and SK. A detailed coding agenda was developed with code and category descriptions to guide the coding process (online supplemental table 3) and distributed to all authors for feedback. Thereafter, categories were identified inductively.

Code labels were lifted directly from participants' quotes, identifying descriptive categories first.[32] Higher level abstract categories were systematically identified separately by MA and SK as coding progressed, tested against the data and consolidated repeatedly. Variations among categories were derived as recommended by Sbaraini et al.[49] The data were broken down, intrinsically compared to explain variations and recombined in an abstract manner describing the relationships.

The line-by-line open coding method was used while the RQDA package in the R software was used for the coding process.[50]

### Patient and public involvement

The circulatory process of qualitative research using grounded theory ensured that patients and physiotherapists as public partners played a key role in theoretical sampling and developing the interview guideline.[40] The use of the SBT was informed by collaborative efforts and consultations of research team members including KS working intensively with patient consultants.

Specifically, patients were consulted in development and tailoring of the information videos and two patients were part of the pilot team for the interview guidelines. They informed the burden of information, time allocated for participation and direction of further enquiry under the grounded theory methodology. Patients' contributions and referrals contributed to the ideas guiding further recruitment. Patients and physiotherapists as public partners helped in the tailoring of the interview questions comprising key questions, maintenance and follow-up questions. This included adapting wording and sentence structures, and including simple patient-specific information. After the interviews, summary notes were read back to patients for confirmation and they gave feedback on the interviews.

### RESULTS

A total of 33 participants were contacted and at different stages by email, and 25 consented to participate. Of the

**Table 3** Physiotherapists' sample description

| Set* | Title† | Age group (years) | Sex | Practice | Experience with LBP | Level of education | Area of interest | Work setting |
|---|---|---|---|---|---|---|---|---|
| 1 | PT 1 | >30–40 | M | >10–20 | >10–20 | MSc | Orthopaedics, private practice | Physiotherapy training institute |
| | PT 2 | >30–40 | M | >0–10 | >0–10 | BSc | Orthopaedics, private practice | Primary health |
| | PT 3 | >20–30 | F | >0–10 | >0–10 | BSc | Women's health | Private practitioner |
| | PT 4 | >30–40 | M | >0–10 | >0–10 | PhD | Orthopaedics, educator | Physiotherapy training institute |
| 2 | PT 5 | >30–40 | M | >0–10 | >0–10 | BSc | Orthopaedics, paediatrics | Teaching hospital |
| | PT 6 | >30–40 | M | >0–10 | >0–10 | BSc | General practice | Specialist hospital |
| | PT 7 | >30–40 | F | >0–10 | >0–10 | BSc | Neurology, paediatrics, geriatrics | Teaching hospital |
| 3 | PT 8 | >30–40 | F | >10–20 | >10–20 | BSc | CRP, orthopaedics | Teaching hospital |
| | PT 9 | >30–40 | M | >0–10 | >0–10 | BSc | Paediatrics | Teaching hospital |
| | PT 10 | >20–30 | F | >0–10 | >0–10 | MSc | Ergonomics, occupational | Corporate organisation |
| 4 | PT 11 | >30–40 | F | >0–10 | >0–10 | BSc | CRP | Teaching hospital |
| | PT 12 | >30–40 | F | >10–20 | >10–20 | MSc | Orthopaedics | Physiotherapy training institute |

*First, second, third and fourth iterative interview rounds.
†Physiotherapists' pseudonymous designation comprising the numerical order of interviews used to reference physiotherapists' quotes.
BSc, Bachelor of Science; CRP, cardiorespiratory physiotherapist; LBP, low back pain; MSc, Master of Science; PhD, Doctor of Philosophy; PT, physiotherapist.

eight remaining participants, two patients were uninterested, while six physiotherapists could not find the time for the study. This resulted in a response rate of 81% for patients and 70% for physiotherapists. All consented participants were interviewed, 14 male and 11 female (12 physiotherapists and 13 patients), lasting an average of 50 min per individual. Physiotherapists had backgrounds in various specialisations and represented a variety of work settings. Patient participants were classified as 'high risk, medium and low' based on the SBT.

Tables 2 and 3 show detailed information on participants' practice experience, experience with LBP, qualification, specialty, work setting and STarT-Back classification.

The perspectives of physiotherapists and patients on the implementation of a tailored stratified care approach were captured in seven main hierarchical categories: (1) recognising the need for change, (2) acceptance of innovation, (3) resistance to change, (4) adapting practice, (5) patient's learning journey, (6) trusting the therapist and (7) needing conviction (table 3). These categories varied based on four themes: (a) tradition of treatment, (b) evolution to a new system, (c) experiences, (d) strategies for implementation (table 4). While some categories focused only on physiotherapists, others focused only on patients, and on both physiotherapists and patients.

## Resistance to change

This is a mixed category as it contains aspects focusing on patients and physiotherapists. It describes the resistance perceived and experienced by physiotherapists and patients concerning the implementation of the stratified care approach. It further describes the physiotherapists' and patients' perception of challenges to the fundamental change and strategies to overcome resistance.

Participants opined that physiotherapists' *tradition of treatment* was a major challenge, as it seemed to be an overwhelming issue sustained by *incentives attached to the usual practice.* Key aspects are ego, self-confidence in their current practice experience and finance. There is a long-held tradition of treatment with non-evidence-based methods, diagnosis and radiographs that the patient has got used to and this tradition remains deeply entrenched in practice.

> You need more patients to have more income, so the more the patients the more the income, so that is the case. (PTm3)

The variation *strategies for implementation* indicates *targeting the attitude of physiotherapists and patients,* which is seen by participants as a major strategy to improve the chances of successful implementation. The physiotherapist and patients see themselves as mostly complacent. The patients blamed the physiotherapists for not being attentive to their needs and the physiotherapists blamed the patients' attitude as a deterrent for being unable to improve their condition.

> I know patient attitude matters…but at least play your role and if the patient misbehaves you know your job

**Table 4** Categories and themes of variation

| Categories | Themes of variation | | | |
| | Tradition of treatment | Evolution to a new system | Experiences | Strategies for implementation |
|---|---|---|---|---|
| Resistance to change | Incentives attached to the usual practice | Overcoming patients' expectations | Organisational culture | Targeting attitudinal change |
| Acceptance of innovation | Ease of transition | Open to new knowledge | Steps towards optimising practice | Work settings |
| Adapting practice | Need of standard for regulating the practice | Cultural adaptations | Use of communication | Awareness for patients and PT |
| Patient's learning journey | Needing a complement to usual care | Recognising unhelpful treatments | Learning to live with pain | Taking charge |
| Trusting the PT | Getting some help | Learning with practice | Therapists doing their best | Cooperating with the PT |
| Needing conviction | Reliance on investigations | Patient education | Self-discovery | Struggle against false information |
| Recognising the need for change | Lack of training to give psychologically informed therapy | Embracing a different approach | No complete relief | The role of funding |

PT, physiotherapist.

is done. If your recommendations are practicable you will get a good attitude from them. (PatM3)

Participants' perceptions varied in relation to an *evolution to a new system* as therapists see *patient expectation* as a primary deterrent to implementing stratified care that needs to be addressed. It feels to them like all their suggestions and efforts to optimise practice will not work because the patients come with a certain expectation. They expect a strong touch, massage or something physical that would bring the pain down to zero. There is, however, an enabling situation seen by therapists and patients that these expectations can be adjusted and overcome.

Then for the patient, … they might not be satisfied because they are expecting more from you…if you don't explain why you have to do that and you just do it they won't feel happy. They feel they need some form of hard touch, strong touch. (PTf4)

From the experiences of physiotherapists and patients, the preponderant *organisational culture* makes treatment less efficient and less effective. They have limitations in their ability to ease the changes beyond a certain point, like organising situations, follow-ups or influencing patients' working conditions due to a systemic resistance.

An example is the government trauma centre I worked in, they had made a move to implement an electronic documentation system for use, but eventually, it never saw the light of day, because the attitude towards it was terrible from the management and recipients. (PTm6)

## Acceptance of innovation
This is a category focusing on physiotherapists. It describes a positive reaction from physiotherapists to the idea of

implementation of stratified care, highlighting their perspectives on the best settings and enabling conditions for implementation with inputs from patients.

Some physiotherapists reported similarity of stratified care to what they already know comprising the usual *tradition of treatment* and practice in their management of LBP, as they have background knowledge of the biopsychosocial approach and various treatments for patients. These are individual aspects of stratified care and participants reported that this *eases the transition* and aids them to adopt the stratified care in its entirety.

I won't say the knowledge of psychosocial problems associated with LBP is low among physiotherapists any longer. I think there has been a lot of awareness and people are getting to know about those issues. We have been always talking of psychosocial aspects and the likes. I think this brings it more to the fore…I think using this approach helps… it's going to be easier for them to incorporate them into routine practice. (PTm5)

Physiotherapists also describe an *evolution to a new system* comprising some form of realisation, that this approach is fundamentally new, uncommon and not used in its entity in clinical practice, hence they *were open to new knowledge* describing stratified care as an improvement to current care and welcome development.

It is very different from what we do here, the treatment here is based on diagnosis, this approach is based on prognosis so yeah, it is different. (PTf4)

From their *experiences*, participants came across situations where they made *steps towards optimising practice*, instances where innovations, ideas or approaches have been introduced in their various work settings, the process that helped and the positive responses from colleagues.

Key aspects were the use of clinical discussions, through physiotherapists' cadre and time-saving measures for the patient. They think the stratified care being introduced in a similar stepwise manner will be successful.

> Many have been introduced to us by different people, I have even tried to introduce one and it was not a big deal if you have a proper literature backup. Once it is adopted everyone used it, for instance, when Kinesio-taping was introduced to us, the same thing happened. it was accepted and then everyone practised using it. (PTf11)

Participants expressed differing perspectives on *strategies for implementation* of stratified care in relation to the most viable *work setting* to target since there exist different situations and ideologies at the private and government establishments. Participants felt it would be more applicable if targeted at the government hospitals since the therapists are paid by the government; a fixed amount and more treatments do not lead to more money.

> Then for physiotherapists, working in government hospitals they might be encouraged to use the approach, but not therapists in the private sector… this approach will reduce their revenue and will not be welcomed. (PTf10)

### Adapting practice
This category applies to patients and physiotherapists. It describes the suggestions on adaptations in terms of creating awareness and modification of the approach to suit the Nigerian context and enhance its use. Here, participants highlight their opinions and experiences on ways to adapt the stratified care approach.

Participants speak of *sociocultural adaptations* relating to language, religion, the culture of respect and hierarchical implementation from the seniors to juniors. This was highlighted as a viable means of adapting practice at the introductory phases in attempting the evolution to a new system.

> Our environment is peculiar things have to go through certain paths, we need to convince the seniors. Then it can be implemented on a departmental level. It is practically impossible for one person to champion it. (PTm9)

Most participants mentioned that the *use of communication* was a major way to inform adaptation and improve the patient–therapist relationship. From their *experiences*, when communication between patient and physiotherapist was used, it was seen to produce positive effects.

> In a way, some part of my treatment helped me psychologically, there are days he would talk to me so I can help myself. (PatF7)

Participants argued that the absence of a standard of practice embedded in their usual *tradition of treatment* and tailored to the needs of Nigerians was a major gap

in practice. This situation led to a clamouring for *standards to guide practice and regulation*. Adherence to such a standard of practice would enhance adaptation for implementation.

> In Nigeria, we do not have a standard way of doing things. what hospital A is doing is different from what Hospital B is doing… everyone is just doing what they want. (PTf7)

Participants confirmed that creating *awareness for patients and physiotherapists* about the approach is a major *strategy for implementation*. While some opined this can be done at undergraduate and postgraduate levels for physiotherapists, patients' awareness was achievable through outreach.

> Patients…should know that physiotherapy involves advice too and they will feel better. it would also help physiotherapists a lot if they are aware of the approach. (PTf4)

### Patient's learning journey
This category focuses on patients and physiotherapists. Here, patients have experienced care from various fronts and have formed their opinions about what they have experienced to be helpful to their condition. This was corroborated by the physiotherapist opinions about the patients' learning experiences from their perspective.

Patients here describe having undergone a learning journey, as they have gathered added experiences. Participants explain how the debilitating condition of the patient's experiences has resulted in personal difficulties. They resolved in their minds to change their outlook, *learn to live with pain* and accept the condition, and this has improved their outlook.

> It affected me psychologically, I was worrying but I have learned to live with it, I think I am better at managing myself …so also, I have learned, I have concluded that no kind of intrusive treatment can work for this my case, that I just need to be careful with how I live my life. (PatM3)

For patients, the usual *tradition of treatment* care was seen to be insufficient and there are aspects that patients feel would *need some form of compliment*. They feel this can be achieved by treatment from traditional bone setters which is seen as a compliment or an alternative.

> It is not everything science can explain…as some hospitals do not have machines, these strong (traditional) hands can be used instead…so some things the hospital cannot handle, these (bone setters) know where and where to touch. (PatF11)

Participants report that over the past recent years, patients have gradually become mentally equipped to *recognise unhelpful treatments* by themselves and develop the ability to differentiate between what is beneficial and what is not. This realisation is seen as a major stride in the

*evolution to a new system* and a viable enabler to the implementation of stratified care.

> When they visit these, they come back (from traditional bone setters' treatments) with complications that I have to deal with so it is a barrier, and over the years it was more, but recently the percentage is coming down now it should be about 30 percent but back then some years ago it was about 70 percent. it makes their treatment longer. (PTf8)

Patients mentioned that they have gathered knowledge of self from various treatment exposures, from self-care and traditional bone setters' treatments. They have now decided that *taking charge* of their treatment is a vital *strategy for implementation* and ultimately a strategy to help themselves.

> My advice is that people should exercise and remain active. I know the positive side of exercise, it is extremely good. That is why I don't miss it at all, no matter what happens. (PatM5)

### Trusting the therapist

This category focuses on patients. It describes the patient's views of the physiotherapist as a dynamic expert with whom they need to give their trust and cooperation based on the efforts of the physiotherapist to help their situation and their experiences with treatments.

Patients agreed that some aspects of the usual *tradition of treatment,* especially those focused on exercise and advice, provided some relief to patients; they reported they *got some help* from it. Even though not permanent, they relate some improvement in their condition and trust that based on this result more can be expected.

> Just the exercise he prescribed, stretching on the bed, but as I was doing it initially I was feeling the pain much but later it reduced, formerly I counted just 20 counts, but now I can count up to 40 without feeling much pain, the physiotherapist gave me the exercise. (PatF9)

Most patients expressed that the physiotherapists will gradually *evolve to the new system* in their practice and that some therapists are already on the right track, *learning to change their approach with practice.* They feel exploring ideas like this through research is needed to evolve more.

> This can be the case sometimes, in a bid to solve a problem they solve others. That's how they learn, they learn by practice. Experience is the best teacher, so most times experience comes into play. (PatM5)

The patients acknowledge that the *physiotherapists are doing their best* by putting in efforts to optimise care in ways they can and they appreciate the efforts. They notice from their personal *experiences* that when some more is done to help their condition, it encourages them to focus more on the efforts.

> Yes, for me, they did the best they could I would say, for the treatment too… they used to tell us and give us hope, the possibility to get better. They have human sympathy, they advise us sometimes that helps. (PatM4)

The patients consider the physiotherapist's role as providing care and support for the patient when needed and their role as patients is not to interfere with the process. As a *strategy for implementation,* they felt they need to *cooperate with the physiotherapist* but largely leaving the aspect of care for the therapists.

> When you know this is what your clinician wants to achieve then anything he tells you would do it, do it so that you can get a good result because when you do not follow up well you might think it did not work because didn't follow instructions well, so if you cooperate you would be able to achieve what you want. (PatF1)

### Needing conviction

This category focuses on patients. It describes the patient's exposure to false information sources and the need for physiotherapists to provide trustworthy education to get their full cooperation. This equips patients on their road to self-discovery and guides them to conviction.

Participants argue that for patients to be able to accept change, they need to be informed properly to a level of conviction. Patients have multiple sources of information and this needs to be presented clearly and reasonably to influencing attitudes and lifestyle. If *patients can be educated* and the procedures explained properly, it will go a long way to influence their expectation, help them become more receptive to a change in approach and a viable enabler in the *evolution to a new system.*

> The patient needs such conviction, he should not leave with the feeling that 'that clinician does not have my interest and doesn't want to treat me let me not waste my time to go back there…. he has to be convinced he is not just assigned to that group because you feel that his problem is not serious enough to be attended to. (PatM2)

Physiotherapists agree that the *tradition of treatment* they are already used to involves *reliance on investigations,* a diagnosis-based approach, using radiographs and passive treatments. Since the patients have become used to that, they need some form of conviction to accept the change. Some patients enjoy passive treatments and would not only prefer that but insist it is the right thing to do.

> The only decision you make that will make sense to me is based on evidence from X-rays, tests and the likes not just based on your opinion. (PatM3)

Patients relate situations when they needed to be convinced from within through some form of *self-discovery,* even though their therapist tried by advising

them they still needed to see for themselves to be successfully convinced to overcome the initial resistance. Based on their *experiences*, they relate that their outlook changed and they experience better outcomes.

> They told me to check my body mass index, but I told them it might not be right, for the past 30 years I have never gone below 80 kilograms, so going to that weight I don't know how I would look like. It would be terrible…I did physiotherapy, took medications and went to hospitals several times. Until I discovered that my weight was the cause… I can manage myself now. (PatM5)

Patients emphasise that there is a variety of information sources and many of which are misleading. These can come from other patients, from the internet or clinicians with certain motives. There is a constant *struggle against wrong information* by the patients which the therapists need to drive against to optimise care. To combat this, information pools and research banks were suggested as viable *strategies for implementation*.

> Yes, although if it is the general internet sometimes it gives false information too. Information pool is needed, as a research bank for publications available to all clinicians so that anyone who needs it can go into such pool in managing their patient. (PatM5)

### Recognising the need for change

This category focuses on physiotherapists and patients. It describes participants' attention to deficiencies and deterrents in physiotherapists' knowledge and practice, further corroborated by the patient's dissatisfaction with the current treatment outcomes which can be enabled by acceptance of the stratified care approach and proper funding.

Physiotherapists admitted to having gaps in knowledge and *lapses in training* that might affect their ability to deliver optimal biopsychosocial care; this has resulted in sticking to the more familiar *tradition of treatment*. This issue of competency was a major setback. Most physiotherapists felt incompetent to handle such an approach especially the prognostic outlook and psychologically informed therapy.

> Physiotherapists in Nigeria do not have the training to give psychologically informed therapy, our training does not cover that, except we just use our instinct, most times what physiotherapists do in the name of psychological therapy is just talk and say some things through instinct. (PTf10)

Many patients have *experienced* treatments involving significant efforts with suboptimal results. Some now seek solutions elsewhere because there has been *no complete relief*. They express disappointment with their situation and frustration with the care they receive. They all feel that things need to change in the way physiotherapists treat patients.

> I had received treatment for some years until 2016, no relief after physiotherapy and I was contemplating travelling abroad for treatment. Clinicians treating patients need the training to update their treatment methods. (PatM3)

Participants complain of issues of infrastructure and patient load. These issues are systemic constraints that are neither the fault of the patients nor the therapists and have a common underlying denominator as a lack of proper funding. Participants jointly suggest improved *funding* to the health sector as a sound *strategy for implementation* aimed at improving services.

> There is a proverb in my place 'good soup cost good money'. So, money plays a bigger role in this. It needs to be available for any of these to work. (PatM5)

Physiotherapists and patients agreed they need to recognise the potential to optimise care to be open *to accepting a different approach*. The potential advantages need to be clear to participants, creating an undeniable appeal of a different approach. This is seen as a facilitator for the *evolution of the new system*.

> Well, I think it is a new approach. It is good, you can then know the kind of treatment and the extent to which you should offer treatment. I believe that this approach goes a long way into what results. It plays a big role. (PatM3)

### DISCUSSION

Exploring the perspectives of physiotherapists and patients with LBP in this study has produced insight into the varied barriers and enablers affecting the implementation of stratified care in Nigeria. Findings from this study show that some physiotherapists described stratified care as a familiar approach to be welcomed. This is consistent with previous knowledge; Sanders *et al* reported thoughtful obedience from physiotherapists[51] and Odole *et al*[52] wrote about the willingness of physiotherapists to adopt innovative care methods. A reason for this could be having previous knowledge from standard entry-level training that encompasses basic components of psychosocial care and evidence-based practice with an emphasis on communication as described by Abaraogu *et al.*[53] Other studies show physiotherapists had previous knowledge of the biopsychosocial approach.[54 55] This shows the viability of adopting a stratified care approach and ease of transition among physiotherapists.

However, for implementation, the physiotherapists in our study described more challenges of implementing the approach than the patients. Two possibilities could explain this: first, a negative attitude towards change; as studies reported, clinicians do not always deliver therapy according to the standard they received during training.[28 56 57] A second possibility could be that they understand the intricacies of use better than the patients.[15]

Research shows that an understanding of the complexities involved in an intervention creates in itself a capacity for implementation.[43 58] Thus, physiotherapists especially expressed concern on internal and external resistance they face in optimising practice including patient's expectations, patient's attitude, patient-load-to-physiotherapist ratio and socioeconomic constraints consistent with findings on the implementation of the stratified care in other contexts.[59–61]

Despite these perceived barriers, from the interviews, some physiotherapists consider the current practice tradition to be outdated, but this treatment tradition is further enabled by their perception of patients' expectations and the physiotherapists' attitude as well. They are concerned that patients may be unsatisfied with their treatments if they change and thus be swayed towards traditional bone setters, and they might lose funds or relevance while adopting a stratified model of care. Studies show that truly, patients could be unappeased with their treatment with stratified care and this can be significantly linked to their expectations.[62] This is consistent with a recent study showing that due to dissatisfaction with their treatment, some patients have gravitated towards treatment by traditional bone setters.[63] From the patient's point of view, it can be derived that ideas on treatment goals, preference for passive treatment, how much treatment and the nature of treatment necessary for someone in their condition are being influenced by their clinicians and have strongly shaped their belief system and expectations. This has a bearing on the attitude of patients towards stratified care and their expectations.[15]

Patients in this study suggested that it is the responsibility of their clinician to provide the conviction needed to secure their cooperation in stratified care implementation. Some believe that if their clinicians properly introduced it to them via education and meaningful communication, it would be seamless. These findings agree with previous research indicating that patients valued healthcare professionals who not only listened to them[64 65] but also took time to explain their conditions and options in lay terms and helped them navigate the healthcare system.[66] This reveals a mismatch in understanding between patients and physiotherapists and an intrinsic strategy for implementation. The patients have expectations, but these are not as strong as the physiotherapists believe and are fuelled by the therapists themselves. Hence, the therapists can modify the patient's expectations strongly by first, being aware of the areas where these differences in perspectives exist. Studies suggest that with further training, experience and learning from open-minded clinical encounter's divergent areas can be identified and managed.[67 68] Second, change in physiotherapists' beliefs, enhanced communication and strides aimed at patient conviction can contribute to modifying patients' expectations.[69]

Another key issue raised as a barrier to implementation is physiotherapist competency in the areas of psychosocial care for high-risk patients. This is a key area of significance paramount in other contexts and among other health practitioners.[61 70] Physiotherapists express concerns about perceived deficiencies in training as seen in previous research.[71 72] Training is required for physiotherapists to deliver a broader biopsychosocial model of care[73] and a key component for successful implementation as shown in research.[43 58] Some question their competence in handling high-risk patients majorly highlighting therapists' need for competency in the psychosocial aspect of patient care. This is consistent with studies that show physiotherapists today do not feel adequately trained to deal with complex high-risk patients.[51 59]

Patients however trust in the physiotherapist in terms of training and competence. This could be due to their experiences because in seeking help, many have resorted to a variety of care from traditional massage and bone setting to the use of devices. Patients relate they have come to trust their therapists and have discovered from experience that being involved in their care, self-management and having an active lifestyle are more beneficial. Such understanding drawn from experience could be a motivation for behaviour change in the direction of implementation.[58] This leaves a new role for physiotherapists, providing enablement for their patients instead of dependency.[59 74] Concordant to this, research reveals that this supporting role should include reducing barriers and increasing the patients' capability to self-manage.[58]

For physiotherapists, major suggestions made were to enforce paradigm shifts in standardising practice, early undergraduate training and workplace mentorship. These interventions support lasting behavioural change[58] and have a bearing on the success or failure of the implementation efforts.[75] Key strategies to implementation are the stepwise introduction of the concept to colleagues especially those of specialty groups relevant to musculoskeletal physiotherapy, emphasising the research results and lobbying the seniors to key in. Some physiotherapists relate that hierarchical implementation has great potential to aid practice among physiotherapists. The driving influence of other clinicians in the higher cadre, senior/supervisors can have a positive impact on the quality of healthcare delivery offered by junior physiotherapists and hence implementation.[76 77]

Suggestions on where implementation should commence were also paramount in the discussions. Physiotherapists relate that implementation efforts should be focused first on the government hospitals.

Regarding the subgroups, physiotherapists practising in federal settings have the opinions that the aspect of high-risk management will not work in federal settings, due to the high patient load as two studies confirm.[78 79] They feel the assessment and delivery of psychosocial treatment will be a problem due to time. They have the opinion it will work better in the private settings, because private sector physiotherapists have fewer patient loads and perhaps more time. This is consistent with the STarT-Back trial where sufficient assessment time of 45 min allocated

to patients in the high-risk category contributed to a successful implementation outcome.[19]

However, physiotherapists in our study felt the low-risk approach will work better because the therapists are paid by the government and they will welcome the idea as it helps to reduce the patient load.[78 79] The private physiotherapists however opined that they need the patients to keep coming, due to financial reasons, hence the low-risk approach will be undesirable. However, they welcome the high-risk approach, as it will improve the quality of care, but the time for assessment will be an issue also as they too have limited time allocated for assessment as research shows.[79 80] Further research reveals that private practice is more satisfactory to patients in terms of interpersonal quality of care, more attentive personnel and a broader choice of provider.[79 81] The patients in our study, similar to participants in a recent review by Lim *et al*, require conviction, quality communication and time for successful implementation.[82]

A strength of this study is its design, adhering to a rigorous method to systematically and meticulously draw out comparisons and variations until saturation was reached. The sampling strategy in this study ensured that physiotherapists and patients in both urban and rural settings were contacted, giving equal opportunity for referral and response. The sample thus included participants of varied sex and age, physiotherapists from different contexts, patients with acute and chronic health problems, patients who visited the physiotherapists and other healthcare institutions, and importantly, our sample comprised patients of all risk groups in the STarT-Back risk classifications. These were checked several times until saturation was reached.

One limitation of this study might be in the presentation of findings. A common challenge with presenting results in grounded theory is the degree of detail in outcomes.[83] However, through constant comparisons, the authors created sufficient connections between the highest levels of abstraction and the data through variations. These were thoroughly explained ensuring all aspects of the data were represented.

This study was carried out in Nigeria, hence care should be taken when generalising results to other countries. However, having context factors in mind, we believe that due to the rigour of the study, interesting comparisons based on these results can be made.[15]

Additionally, the researchers' own subjectivity, beliefs and experiences might have played an important role. This is not necessarily a limitation, but can be seen as an inevitable and integral part of using grounded theory compared with other qualitative methods such as content analysis.[84 85] A challenge that was overcome during this study was getting the participants motivated to engage in uninterrupted productive discussions. This was achieved by emphasising the relevance of the interviews early during the recruitment process, keeping in mind that stakeholders are often interested in knowledge co-creation and mobilisation when the relevance is succinct.[86]

Thus, comprehensive and rich interviews were achieved with participants in a comfortable atmosphere.

## CONCLUSION

Perspectives of patients about stratified care differ from that of physiotherapists, and both should be aware of divergent areas needing resolution. Emphasis should thus be placed on quality communication among stakeholders. Such communication requires skill on the part of the physiotherapist and sufficient time to deliver, both of which are considered barriers to implementation of the intervention.

Other meaningful barriers identified in this study include the treatment expectations held by patients regarding the method and success of intervention and the incentivised attachment to the usual practice provided by physiotherapists such as using passive methods.

This study further reveals the use of facilitators like patients' trust, specialty groups and contextual adaptations has the potential to aid the implementation of stratified care. To integrate this into clinical practice, a consensus is required from the physiotherapists on key strategies for tailoring these findings, identifying and prioritising approaches to use these facilitators, taking care to overcome the barriers.

**Acknowledgements** The authors express thanks to Faith Ikpami and Ogbulie Smart for their help to recruit patients; Jonathan Hill and Zipporah Echigeme for their input on the introductory materials; Anneke Wolf for her ideas and help setting up the online survey; and Cedric Bender for his ideas on analysis.

**Contributors** MA designed the study; recruited participants; organised data collection; transcribed, analysed and interpreted the data; and drafted the manuscript. JS designed the study, analysed and interpreted the data, and critically revised the manuscript. KS designed the study, analysed and interpreted the data, and critically revised the manuscript. CEM designed the study, recruited participants and critically revised the manuscript. SK designed the study, analysed and interpreted the data, drafted the manuscript and critically revised the document. MA conceived and supervised all aspects of the project, therefore responsible for the overall content as the guarantor.

**Funding** MA received personal funding from the German Academic Exchange Service (DAAD) and Petroleum Technology Development Fund (PTDF) for his doctoral project.

**Competing interests** None declared.

**Patient and public involvement** Patients and/or the public were involved in the design, or conduct, or reporting, or dissemination plans of this research. Refer to the Methods section for further details.

**Patient consent for publication** Not required.

**Ethics approval** Ethical approval was obtained from the Trier University of Applied Sciences and the Obafemi Awolowo University Teaching Hospital, Ile-Ife Nigeria (registration ID: IRB/IEC/0004553). Approval for telephone interviews was also obtained from the Trier University of Applied Sciences. All participants were sent information sheets and provided written informed consent before participation.

**Provenance and peer review** Not commissioned; externally peer reviewed.

**Data availability statement** All data relevant to the study are included in the article or uploaded as supplemental information.

of the translations (including but not limited to local regulations, clinical guidelines, terminology, drug names and drug dosages), and is not responsible for any error and/or omissions arising from translation and adaptation or otherwise.

**ORCID iD**
Mishael Adje http://orcid.org/0000-0002-7631-0588

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
