## [Reviewer comments · BMJ Open]

ARTICLE DETAILS

TITLE (PROVISIONAL)	Patients' and physiotherapists' perspectives of implementing a tailored stratified treatment approach for low back pain in Nigeria: a qualitative study.
AUTHORS	Adje, Mishael; Steinhäuser, Jost; Stevenson, Kay; Mbada, Chidozie Emmanuel; Karstens, Sven

VERSION 1 – REVIEW

REVIEWER	Enthoven, Paul Linkopings universitet, Department of Health, Medicine and Caring Sciences
REVIEW RETURNED	28-Dec-2021

GENERAL COMMENTS	This is a qualitative interview study describing Patients' and physiotherapists' perspectives of implementing a tailored stratified treatment approach for low back pain in Nigeria. The interview questions were drafted in the backdrop of categories in the Consolidated Framework for Advancing Implementation Research (CFIR). For analysis of the interview material grounded theory (GT) is used. GT is a method appropriate to develop a theory. In the current study the theory is already based on the Consolidated Framework for Advancing Implementation Research (CFIR). Therefore, it is not appropriate to apply GT. Although the categories and subcategories look interesting, I would suggest producing these using e.g. qualitative content analysis (ref: Corbin and Strauss, 1996, Basics of qualitative research. Techniques and procedures for developing grounded theory.; Glaser and Strauss 1967: The purpose of grounded theory is to construct theory grounded in data).
--

REVIEWER	Zemková, Erika Comenius University in Bratislava
REVIEW RETURNED	27-Jan-2022

GENERAL COMMENTS	The study investigates the perceptions of physiotherapists and patients in Nigeria towards stratified care for the treatment of low back pain, exploring barriers and enablers to implementation. Findings of this study are in my opinion of relevance to medical research and would fit the scope of the journal. There are, however minor concerns which should be addressed. Abstract: It does reflect the content of the article.
---

	Methods: The study design is appropriate. Procedures are clearly described. Please, specify inclusion and exclusion criteria for participants to be allocated to the study. Page 6, lines 23-24: Participants were contacted following set inclusion criteria specifically comprising physiotherapists and patients living in Nigeria. Results: Findings are clearly presented. Discussion / Conclusion: The discussion reflects what authors found and how it relates to the literature. The authors incorporated previous research into their interpretation of the results. However, relevant limitations of this research should be discussed. Page 21: A strength of this study is its design, ...
--	---

VERSION 1 – AUTHOR RESPONSE

Reviewer: 1

Comments to the Author:

1) This is a qualitative interview study describing Patients' and physiotherapists' perspectives of implementing a tailored stratified treatment approach for low back pain in Nigeria. The interview questions were drafted in the backdrop of categories in the Consolidated Framework for Advancing Implementation Research (CFIR). For analysis of the interview material grounded theory (GT) is used. GT is a method appropriate to develop a theory. In the current study the theory is already based on the Consolidated Framework for Advancing Implementation Research (CFIR). Therefore, it is not appropriate to apply GT.

RESPONSE:

The authors thank Reviewer 1 for these comments.

In our perspective, our use of the ambiguous sentence here might have led Reviewer 1 to a wrong assumption.

'The interview questions drafted in the backdrop of categories in the CFIR' does NOT mean 'the theory was already based on the CFIR' as Reviewer 1 assumed. It rather means the CFIR framework was used to reflect the INITIAL interview questions for comprehensiveness in all aspects relating to implementation.

This is backed up by the following literature: Grounded theory begins with 'Open questions' (Sbaraini et al. 2011) and researchers MUST develop these from existing research as confirmed by Foley et al. 2021: *'The interview guide can and should be informed by the literature on the area of the inquiry... both the interview guide and the literature that informs it should be viewed as a launch pad, not a straitjacket'*. (Reference: Sbaraini et al. 2011: How to do a grounded theory study in BMC; Foley et al. 2021. Interviewing as a Vehicle for Theoretical Sampling in Grounded Theory. International Journal of Qualitative Methods)

The interview questions in Grounded theory should also reflect the depth and span of enquiry as inferred from research by, Willig et al. 2008 and Corbin et al. 1990 *'...The initial*

questions should focus the attention of participants on the phenomenon they want to investigate' taking all aspects into consideration. This was further reiterated by Charmaz et al. 2006: '...plan to gather sufficient data to fit your task and to give you as full a picture of the topic as possible within the parameters of this task'. (Reference: Charmaz K: Constructing Grounded Theory: A Practical Guide through Qualitative Analysis London: Sage; 2006; Willig et al. 2008: The SAGE Handbook of Qualitative Research in Psychology, United Kingdom. SAGE Publications; Corbin et al. 1990: Grounded theory research: Procedures, canons, and evaluative criteria In Qualitative Sociology)

Additionally, this initial interview guide was also modified severally in the direction of the data collected (see Appendix 2) consistent with Grounded theory. It was also flexible, semi-structured and 'theoretical sampling' was employed in subsequent rounds fully compliant with Grounded theory methodology. Hence, an authentic outcome resulted.

Our interview questions development was also in fulfilment of the COREQ criteria number 17 (Tong et al. 2007).

Therefore, by 'Backdrop' the authors meant they had considered the implementation aspect from a holistic viewpoint. We understand that this might have caused a misunderstanding. We have thus replaced this in the manuscript with the following statement in PG6, LINE 26:

'In developing the initial interview questions, the aspect of implementation was duly considered (42)'

- 2) Although the categories and subcategories look interesting, I would suggest producing these using e.g. qualitative content analysis (ref: Corbin and Strauss, 1996, Basics of qualitative research. Techniques and procedures for developing grounded theory.; Glaser and Strauss 1967: The purpose of grounded theory is to construct theory grounded in data).

RESPONSE:

The authors thank Reviewer 1 for these comments.

The researchers believe that content analysis might not be a suitable method in this study and does not fit the criteria by Mayring 2000 '*...The procedures of qualitative content analysis seem less appropriate, if the research question is highly open-ended, explorative, variable..., or if a more holistic analysis is planned.*' (Mayring 2000. Qualitative Content Analysis).

Since our research enquiry was highly explorative, open-ended and variable; aiming to consider all aspects in a holistic viewpoint. Also, the direction of research enquiry was modified via theoretical sampling along with the data collected. We thus consider Grounded Theory more suitable.

The authors, however, also considered that other qualitative methods might give the chance for a more concise presentation of findings in this study. Hence, we added two statements on this in our limitations.

Limitations, PG20, Line25:

'One limitation of this study might be in the presentation of findings. A common challenge with presenting results in Grounded theory is the degree of detail in outcomes (Backman and Kyngas 1999). However, through constant comparisons, the authors created sufficient connections between the highest levels of abstraction and the data through variations. These were thoroughly explained ensuring all aspects of the data were represented'

'This study was carried out in Nigeria, hence care should be taken when generalising results to other countries. However, having context factors in mind we believe that due to the rigour of the study, interesting comparisons based on these results can be made (Sowden et al, 2018).'

Reviewer: 2

Comments to the Author:

The study investigates the perceptions of physiotherapists and patients in Nigeria towards stratified care for the treatment of low back pain, exploring barriers and enablers to implementation. Findings of this study are in my opinion of relevance to medical research and would fit the scope of the journal. There are, however minor concerns which should be addressed.

RESPONSE:

The authors thank Reviewer 2 for the comments on the relevance and suitability of our study.

Abstract:

It does reflect the content of the article.

RESPONSE:

Thanks for this comment.

Methods:

The study design is appropriate.

Procedures are clearly described.

RESPONSE:

Thanks for this comment

Please, specify inclusion and exclusion criteria for participants to be allocated to the study.

Page 6, lines 23-24: Participants were contacted following set inclusion criteria specifically comprising physiotherapists and patients living in Nigeria.

RESPONSE:

Thanks for this important comment. A new statement checked against the inclusion documents was added to the manuscript.

PG5, Line 25: 'To be included in this study, physiotherapy participants had to be licensed and registered by the regulating body (Medical Rehabilitation Therapist Board) of Nigeria and practising physiotherapy in Nigeria with entry-level qualification and above. They should have worked on the STarT-Back information on the introductory video provided. Patients participants had to be diagnosed as having non-specific low back pain by their physician or physiotherapist and have visited a health care institution for physiotherapy care. They should speak and understand the English language fluently, and should be able to fill the STarT-Back questionnaire. All participants must be above 18 years'.

'Patients with a diagnosis or having any signs or symptoms pointing to a severe disease condition (e.g. cancer) as the cause of their low back pain were excluded from this study.'

Results:

Findings are clearly presented.

RESPONSE:

Thanks for this comment.

Discussion / Conclusion:

The discussion reflects what authors found and how it relates to the literature. The authors incorporated previous research into their interpretation of the results.

RESPONSE:

Thanks for this comment.

However, relevant limitations of this research should be discussed.

Page 21: A strength of this study is its design, ...,

RESPONSE:

Thanks for your comment. The following limitations have been added:

Limitations, PG20, Line25:

'One limitation of this study might be in the presentation of findings. A common challenge with presenting results in Grounded theory is the degree of detail in outcomes (Backman and Kyngas 1999). However, through constant comparisons the authors created sufficient connection between the highest levels of abstraction and the data through variations. These were thoroughly explained ensuring all aspects of the data were represented'.

'This study was carried out in Nigeria, hence care should be taken when generalising results to other countries. However, having context factors in mind we believe that due to the rigour of the study, interesting comparisons based on these results can be made (Sowden et al, 2018)'.

'Additionally, the researchers' own subjectivities, beliefs and experiences might have played an important role. This is not necessarily a limitation, but can be seen as an inevitable and integral part of using grounded theory compared with other qualitative methods such as content analysis (Holstein et al. 1997, Strübing 2004)'.

VERSION 2 – REVIEW

REVIEWER	Enthoven, Paul Linköpings universitet, Department of Health, Medicine and Caring Sciences
REVIEW RETURNED	15-May-2022

GENERAL COMMENTS	This is an interesting qualitative interview study that aimed to explore patients' and physiotherapists' perspectives of implementing a tailored stratified treatment approach for low back pain in Nigeria. The authors have made some important improvements in the manuscript. The introduction provides a clear background of the study. The authors have clarified in the text their choice of study design and clarified why they used grounded theory as their method for exploring the topic. The procedures are sufficiently and clearly described. The findings are clearly represented. The discussion reflects the findings and how they relate to the literature. The authors incorporated previous research into their interpretation of the results. The study strengths and limitations are clearly described. The English language can be improved at a few places.
--